# Deciphering the Long Non-Coding RNAs and MicroRNAs Coregulation Networks in Ovarian Cancer Development: An Overview

**DOI:** 10.3390/cells10061407

**Published:** 2021-06-06

**Authors:** César López-Camarillo, Erika Ruíz-García, Yarely M. Salinas-Vera, Macrina B. Silva-Cázares, Olga N. Hernández-de la Cruz, Laurence A. Marchat, Dolores Gallardo-Rincón

**Affiliations:** 1Posgrado en Ciencias Genómicas, Universidad Autónoma de la Ciudad de México, 03100 CDMX, Mexico; ediacara79@yahoo.com.mx; 2Grupo de Investigación en Cáncer de Ovario y Endometrio, Instituto Nacional de Cancerología, 14080 CDMX, Mexico; betzabe100@yahoo.com.mx (E.R.-G.); dgallardorincon@gmail.com (D.G.-R.); 3Laboratorio de Medicina Traslacional y Departamento de Tumores Gastrointestinales, Instituto Nacional de Cancerología, 14080 CDMX, Mexico; 4Departamento de Bioquímica, CINVESTAV-IPN, 07360 CDMX, Mexico; yarely.vera.ms@gmail.com; 5Coordinación Académica Región Altiplano, Universidad Autónoma de San Luis Potosí, 78700 San Luis Potosí, Mexico; macrina.silva@uaslp.mx; 6Programa en Biomedicina Molecular y Red de Biotecnología, Instituto Politécnico Nacional, 07340 CDMX, Mexico; lmarchat@ipn.mx

**Keywords:** long non-coding RNAs, microRNAs, co-regulation networks, gene expression, ovarian cancer

## Abstract

Non-coding RNAs are emergent elements from the genome, which do not encode for proteins but have relevant cellular functions impacting almost all the physiological processes occurring in eukaryotic cells. In particular, microRNAs and long non-coding RNAs (lncRNAs) are a new class of small RNAs transcribed from the genome, which modulate the expression of specific genes at transcriptional and posttranscriptional levels, thus adding a new regulatory layer in the flux of genetic information. In cancer cells, the miRNAs and lncRNAs interactions with its target genes and functional pathways are deregulated as a consequence of epigenetic and genetic alterations occurring during tumorigenesis. In this review, we summarize the actual knowledge on the interplay of lncRNAs with its cognate miRNAs and mRNAs pairs, which interact in coregulatory networks with a particular emphasis on the mechanisms underlying its oncogenic behavior in ovarian cancer. Specifically, we reviewed here the evidences unraveling the relevant roles of lncRNAs/miRNAs pairs in altered regulation of cell migration, angiogenesis, therapy resistance, and Warburg effect. Finally, we also discussed its potential clinical implications in ovarian cancer and related endocrine disease therapies.

## 1. Introduction

Ovarian, cervical, and uterine cancers are the most frequent malignancies of the female genital tract, which represent a serious public health problem worldwide [1]. Of particular interest, ovarian cancer is the most common and lethal gynecological cancer around the world with 295,414 estimated new cases and 184,799 deaths recently reported worldwide [2]. Ovarian cancer comprises a wide range of subtypes, which has been classified into epithelial ovarian cancers (90% of cases), germ cell tumors (5%), and sex cord-stromal tumors (2–5%). It can be stratified in five histotypes namely as low/high-grade serous carcinoma (68–71%), endometrioid (9–11%), clear-cell (13%), and mucinous (3%) [3]. The aforementioned ovarian cancer subtypes showed distinct genetic programs and mutational profiles associated with alterations in the regulated expression of oncogenes and tumor suppressor genes which led to the activation of signaling transduction pathways governing tumor growth, disease progression, and clinical outcomes [4]. Remarkably, the deep molecular characterization of ovarian tumors has revealed that the epithelial ovarian cancer subtype can be divided into two groups dubbed as low-grade serous (type I) and high-grade serous (type II) [3]. Type II carcinoma is more aggressive and associated with progressive disease resulting in most deaths in patients, whereas the type I tumors are less aggressive. The treatment of ovarian cancer is based on surgery followed by chemotherapy using cisplatin/carboplatin and taxanes that represent the first-line agents for patients with advanced disease [5]. However, despite the initial good response to therapy, one fourth of patients progress to recurrent disease with a low overall 5-year survival. In the last two decades, cell biology and genome-wide profiling studies have contributed to the understanding of ovarian cancer disease. In addition, special attention has gained the identification and functional characterization of long and small non-coding RNAs (ncRNAs) as they regulate a plethora of protein-encoding genes related to development and progression of human cancer of regulatory non coding RNAs: lncRNAs and microRNAs.

Non-coding RNAs (ncRNAs) comprise a rich and abundant diversity of RNA molecules in eukaryotic cells, which were originally defined as transcripts without information for protein synthesis. Currently, there is increasing scientific evidence showing that different types of ncRNAs are involved in the regulation of multiple fundamental processes for life, such as cell differentiation, cell division and proliferation, cell cycle, cell death, metabolism, etc. Here, we will focus on the description of the roles of long non-coding RNAs (lncRNAs) and microRNAs as the main types of deregulated ncRNAs with a relevant role in carcinogenesis. In particular, we highlighted the description of functional interactions between lncRNAs and microRNAs which interact with each other in large regulatory networks that better describe the ovarian cancer cells behavior and tumorigenesis mechanisms, as they may influence gene expression at the transcriptional and posttranscriptional level. In ovarian cancer, a number of individual studies reporting on lncRNAs and microRNAs genomic profiling have been published, but frequently the information remains dispersed in multiple datasets such as Gene Expression Omnibus (GEO) and The Cancer Genome Atlas (TCGA), thus in this review we aim to put together the omics data and individual reports on lncRNAs/microRNAs pairs published in diverse studies for better reference and understanding for cancer researchers.

## 2. The Expanding Repertoire of Regulatory LncRNAs and MicroRNAs

LncRNAs represent a class of ncRNAs longer than 200 nucleotides that are not translated to proteins. LncRNAs are the most abundant class of ncRNAs and have been proposed to carry out diverse cellular functions including epigenetic regulation, transcriptional modulation in cis or trans, organization of nuclear domains, post-transcriptional processes, and regulation of proteins or RNA molecules to maintain cellular homeostasis [6]. More than 55,000 lncRNAs genes are currently listed on LNCipedia 5.2, a database specialized in human lncRNA, of which just over 1500 have annotated with functional information [7]. According their location in genome, lncRNAs are classified as sense, antisense, bidirectional, intergenic, and intronic, while regarding their subcellular location they are grouped as nuclear and cytoplasmic lncRNAs [8]. Nuclear lncRNAs are epigenetic regulators of gene expression functioning as: (i) RNA decoys, by sequestering transcription factors and other proteins from chromatin and balancing their availability to drive gene transcription [9]; (ii) RNA structural scaffolds, working as platforms for the recruitment of proteins bringing in proximity to specific sites, and to facilitate the assembly of large ribonucleoprotein (RNP) complexes with regulatory functions [10]; and (iii) enhancer RNAs, influencing the formation of chromatin loops anchors and topological three-dimensional (3D) organization of genome [11]. On the other hand, cytoplasmic lncRNAs may comprise lncRNAs transcribed from nuclear DNA or from mitochondrial DNA. Cytoplasmic lncRNAs also interact with RNA-binding proteins (RBPs), forming lncRNA-associated ribonucleoprotein complexes (lncRNPs) [12]. Cytoplasmic lncRNAs are regulators of gene expression that function as: (i) RNA decoys, by sequestering RBPs, thus influencing the abundance and translation of mRNAs, they can also sequester miRNAs, functioning as competing endogenous (ce)RNAs [13,14]; (ii) modulation of signaling pathways, through the binding of RBP to lncRNA promoting conformational changes that activate signaling molecules and/or the alteration of their phosphorylation state [15], and (iii) regulate protein stability by controlling degradation pathways, in particular by enhancing access to the ubiquitin/proteasome machinery [16,17]. Recently, aberrant expression of lncRNAs have been identified in human malignancies [18], where they can function either as oncogenes and tumor suppressors, and their deregulation is associated with increased cell proliferation, tumor growth, angiogenesis, invasion and metastasis, drug resistance, resistance to apoptosis, and clinical poor prognosis [19]. Dysregulation of lncRNAs has been implicated in glioblastoma [20], breast cancer [21], colorectal cancer [22], hepatocellular carcinoma [23], and leukemia [24], etc.

MicroRNAs are evolutionary conserved tiny non-coding single-stranded RNAs of 21–25 nucleotides in length, which negatively regulate the gene expression at posttranscriptional level. MicroRNAs contain a seed region (nucleotides 2–7) which pair to conserved sites in the 3’ untranslated regions (UTRs) of target mRNAs to activate the post-transcriptional repression [25]. A recent study [26] estimates that human miRNome could consist of around 2300 different mature miRNAs, which would be responsible for controlling about 50% of genes encoding proteins [25]. Increasing evidence shows that microRNAs control key genes and cellular pathways related to cell proliferation, apoptosis, invasion and metastasis, cancer stem cells, angiogenesis, and drug resistance. A plethora of reports showed that microRNAs and lncRNAs are pivotal in carcinogenesis, thus they represent potential biomarkers and therapeutics targets. In recent years, the use of new high-throughput RNA sequencing techniques has allowed the identification of lncRNAs and microRNAs transcribed from the genome, and the establishment of intricate lncRNAs/microRNAs coregulation networks with outstanding relevance in carcinogenesis. The development of computational tools and experimental strategies that define the intricate connections existing in regulatory networks, and contributing to their functional characterization, becomes necessary (Figure 1).

## 3. LncRNAs/MicroRNAs Pairs Controlling Cell Migration, Invasion, and Metastasis in Ovarian Cancer

Cell migration, invasion, and metastasis are relevant cancer hallmarks as they are responsible for tumor cells spreading in the body, resulting in the formation of secondary tumors and high risk of death. Remarkably, the epithelial-mesenchymal transition (EMT) is an essential step which contributes to the dissemination of cancer cells and formation of distant tumors. This phenomenon occurs when tumor cells lose their epithelial features which allowed leaving the primary focus, invading neighboring tissues and enters the blood circulation to reach distant tissues and organs [27]. Therefore, the acquisition of invasiveness is associated with the loss of epithelial features and a gain of a mesenchymal phenotype. An increasing number of lncRNAs and microRNAs have been implicated in the regulation of EMT, invasion, and metastases [28]. Therefore, the knowledge of the roles of ncRNAs in cell migration and metastasis could help in the discovery of novel biomarkers and potential therapeutic targets in ovarian cancer. Next, we reviewed the best-characterized lncRNAs/microRNAs pairs involved in cell migration and invasion regulation in ovarian cancer (Figure 2). For clarity, we have included the description of lncRNAs in which, in our opinion, the studies have a relevant interplay with microRNAs impacting the development and progression of ovarian cancer.

### 3.1. HOX Transcript Antisense RNA (HOTAIR)

One of the first oncogenic lncRNAs identified was HOX transcript antisense RNA (HOTAIR), a polyadenylated ncRNA transcribed from the antisense strand of the HOXC genomic cluster [29]. In normal cells, HOTAIR has key epigenetics functions as modulator of chromatin states by recruiting the polycomb repressive complex 2 (PRC2) and influencing its occupancy at gene promoters, thus regulating the gene expression [30]. HOTAIR, as many other lncRNAs, modulates the microRNAs-mediated suppression of gene targets by sponging microRNAs with tumor suppressor functions, enhancing the malignant behavior of tumors [31]. Three studies documented the roles of HOTAIR in ovarian cancer. In an early report, Dong et al. found that the expression levels of HOTAIR were increased in SKOV3 and OVCAR3 ovarian cancer cells. In addition, the expression of PIK3R3 protein, a member of the phosphoinositide-3-kinase (PI3K) family of proteins, was upregulated in tumor tissues. After knockdown of HOTAIR, the protein and mRNA levels of PIK3R3 were decreased, and vice versa, when PIK3R3 was silenced, the levels of HOTAIR were decreased suggesting that HOTAIR may interact with PIK3R3. The expression of both HOTAIR and PIK3R3 was downregulated when ovarian cancer cells were transfected with miR-214 or miR-217 mimics. Moreover, silencing of HOTAIR or PIK3R3 resulted in decreased cell proliferation and invasion in SKOV3 cells. These data suggested that HOTAIR interacts with PIK3R3 to regulate cell proliferation and invasion through miR-214 and miR-217 [32]. However, HOTAIR expression in ovarian cancer tissues was not reported in this first study.

In the same year, another study from Zhang et al. confirmed the significant upregulation of HOTAIR in a cohort of ovarian cancer patients, which was positively associated with metastasis. The ectopic upregulation of HOTAIR augmented the cell proliferation and migration abilities of HOTAIR-deficient non-tumorigenic HeyC2 ovarian cancer cells. Contrarily, the silencing of HOTAIR activated apoptosis and suppressed cell invasion. Mechanistically, HOTAIR acted as a competitive endogenous RNA (ceRNA) by sponging the miR-373 which in turn regulates the expression of Rab22a, in consequence, the ectopic expression of miR-373 significantly downregulated the levels of Rab22a protein, a small GTPase involved in EMT and cell migration [33].

In the third study, Chang et al. also confirmed the upregulation of HOTAIR in a cohort of ovarian cancer patients. They found that the inhibition of HOTAIR resulted in low cell viability, proliferation, and invasion in vitro. Functional assays showed that both CCND1 (Cyclin D1) and CCND2 (Cyclin D2) were targeted by miR-206 which in turn was sponged by HOTAIR. Congruently, the inhibition of HOTAIR resulted in high expression of miR-206 and reduced levels of both cyclins indicating a role for the HOTAIR/miR-206/Cyclin B axis in ovarian cancer [34]. In conclusion, these early studies suggested that the upregulation of HOTAIR contributed to the malignant progression of ovarian cancer.

### 3.2. Metastasis-Associated Lung Carcinoma Transcript 1 (MALAT1)

MALAT1 (also known as NEAT2) is an evolutionary conserved lncRNA overexpressed in human cancers [35]. MALAT1 was found localized into nuclear speckles, a nuclear domain enriched in pre-mRNA splicing factors, where it interacts with serine- and arginine-rich (SR) proteins involved in splicing regulation, and with epigenetic proteins such as the nuclear methyltransferase-like protein 16 (METTL16) [36]. Interestingly, the Drosha-DGCR8 microprocessor complex interacts also with MALAT1 regulating its stability [37], likewise, it was reported that the depletion of AGO2 also produced a stabilizing effect of transcript. The oncogenic functions of MALAT1 include: (i) The regulation of gene expression at posttranscriptional level by regulating splicing of pre-mRNAs into nuclear speckles; (ii) epigenetic regulation of gene expression of target genes by interacting with PRC2 components EZH2, SUZ12, and EED and promoting trimethylation of histone H3 at lysine 27 (H3K27me3); (iii) acting as a sponge of tumor suppressor microRNAs; and (iv) at transcriptional level by interacting with phosphorylated RNA polymerase [36]. Surprisingly, it has been observed that MALAT1 can also function as a tumor suppressor in breast and colorectal tumors [38]. MALAT1 contributes to the aggressive behavior and worst prognosis of ovarian cancer by sequestering diverse microRNAs. For instance, Lei et al. reported that MALAT1 expression was upregulated in ovarian cancer tissues and cell lines. The silencing of MALAT1 resulted in low cell proliferation of SKOV3 and A2780 cancer cells. In addition, miR-506 was downregulated in ovarian cancer tissues and cell lines and negatively correlated with MALAT1 expression. Moreover, miR-506 restoration inhibited DNA synthesis, cell growth, and the expression of RelA-associated inhibitor (iASPP). Mechanistically, these associations were confirmed by direct binding of miR-506 to MALAT1 and the binding of miR-506 to iASPP 3’UTR [39].

A similar study confirms the upregulation of MALAT1 in ovarian cancer cell lines, and tumor tissues which was associated with poor prognosis [40]. Lin et al. reported that MALAT1 was upregulated in ovarian tumors and correlated with overall survival and progression-free survival of patients. Knockdown of MALAT1 decreased cell viability, migration, and invasion. Furthermore, silencing of MALAT1 derepressed miR-143-3p resulting in the inhibition of CMPK [41]. These studies suggested a role for MALAT1 in tumorigenesis and it may be a useful therapeutic target in ovarian cancer.

### 3.3. Plasmacytoma Variant Translocation 1 (PVT1)

A genomic region denoted as variant (6;15) was first discovered as frequently translocated in mouse models of plasmacytoma. The variant (6;15) translocation joined the myc oncogene-bearing band of chromosome 15 and the immunoglobulin kappa band of chromosome 6 [42]. Unexpectedly, a lncRNA dubbed as plasmacytoma variant translocation 1 (PVT1) was found in this region. PVT1 exhibits oncogenic roles and poor clinical implications in non-small cell lung, cervical, breast, gastric, and colorectal cancers [43]. PVT1 was upregulated in breast cancer and has the ability to activate beta-catenin signaling by targeting KLF5 [44]. PVT1 regulates tumor growth by interacting with the histone methyltransferase enhancer of zeste homolog 2 (EZH2) to function as a repressor of p15 and p16 via trimethylating lysine 27 on histone H3 in gastric cancer [45]. In addition, PVT1 knockdown sensitizes cancer cells to radiotherapy by activation of apoptosis and DNA repair in nasopharyngeal carcinoma [46]. In ovarian cancer, two studies reported on the relevance of PVT1 in tumorigenesis. Yang et al. showed that PVT1 was overexpressed in ovarian tumors relative to normal tissues, and patients with higher PVT1 expression have worse progression-free and overall survivals. Furthermore, PVT1 silencing inhibited cell proliferation, and diminished the migration and invasion abilities of SKOV3 cells through the negative regulation of miR-133a [47]. One year later, Ding et al. mined the TCGA database to search for gene copy number alterations in ovarian cancer patients, and found that the PVT1 genomic region showed a high degree of amplification which was correlated with poor survival [48]. Functional assays showed that the silencing of PVT1 resulted in low cell viability, decreased metabolic activity, and cell cycle arrest in A2780 and SKOV3 cells. PVT1 directly binds to miR-140 acting as a sponge. Finally, the binding of FOXO4 transcription factor on TGTTT motif from PVT1 promoter was reported [42,48].

### 3.4. Potassium Voltage-Gated Channel Sub-Family Q Member 1 Opposite Strand Transcript 1 (KCNQ1OT1)

LncRNA-KCNQ1OT1 is located at the KCNQ1 locus in chromosome 11p15.5 in a genomic region containing several imprinted genes. The main functions of KCNQ1OT1 have been reported on the epigenetic regulation of multiple genes via chromatin domain interactions, in a similar way as reported for other lncRNAs such as Xist, HOTAIR, and Airn. KCNQ1OT1 silences multiple genes in the KCNQ1 domain by establishing a repressive and lineage-specific high order of chromatin structure through the recruitment of chromatin and DNA-modifying proteins [49]. It was reported that the KCNQ1OT1 deregulation is involved in the development of diverse pathologies such as diabetic nephropathy, acute myocardial infarction, atherosclerosis, and cerebral ischemia-reperfusion injury [50]. In cancer, it exhibits oncogenic roles in osteosarcoma, hepatocellular carcinoma, bladder, breast, tongue, lung, prostate, and colon cancer [51,52]. Two reports highlight the role of KCNQ1OT1 in ovarian cancer. Liu et al. showed that KCNQ1OT1 promotes ovarian cancer via the miR-142-5p/CAPN10 axis. KCNQ1OT1 expression levels were upregulated in SKOV3 and OVCAR3 cells and associated with worst overall survival of patients. Gain and loss-of-function assays showed that overexpression of KCNQ1OT1 promotes cell proliferation and invasion, whereas its knockdown resulted in the inhibition of these processes. KCNQ1OT1 sponges to miR-142-5p, which in turn binds to calpain 10 (CAPN10) transcript. Moreover, miR-142-5p was found downregulated, while CAPN10 was overexpressed in ovarian cancer cells. Furthermore, the forced expression of miR-142-5p inhibited cell proliferation, colony formation, and invasion [53].

In other study, Lu et al. also showed that KCNQ1OT1 was upregulated in ovarian cancer tissues and cell lines, and associated with advanced disease and poor prognosis in ovarian cancer patients. The specific knockdown of KCNQ1OT1 significantly repressed cell growth, migration, and invasion and activates apoptosis, whereas its overexpression produced the opposite effects. In addition, it was found that miR-212-3p expression was negatively associated with KCNQ1OT1 and that miR-212-3p negatively regulated the expression of lipocalin2 (LCN2) via binding to its 3’-UTR. This study indicates that KCNQ1OT1 stimulates ovarian cancer by modulating the miR-212-3p/LCN2 axis, thus it may be a potential therapeutic target [54].

### 3.5. Growth Arrest-Specific 5 (GAS5)

The specific transcript 5 of the siRNA growth arrest (GAS5), is located on chromosome 1q25 and its length is around ~630 nucleotides [55]. LncRNA-GAS5 is a tumor suppressor in several types of cancer, including breast and lung [56]. In particular, in ovarian cancer it has been shown that repression of GAS5 promotes cell proliferation, migration, and invasion and correlates with poor prognosis in patients. Ma et al. showed that GAS5 promotes ovarian cancer progression via modulating the miR-21/SPRY2 axis. GAS5 expression levels were downregulated in tissues and cell lines of ovarian cancer and associated with advanced clinical stage. Overexpression of GAS5 inhibited the proliferation of ovarian cancer cells, resulting in decreased expression of miR-21 and increased SPRY2. GAS5 sponges to miR-21 which in turn binds and represses the sprouty homolog 2 (SPRY2) transcript. Moreover, miR-21 was found overexpressed, while SPRY2 was found downregulated in ovarian cancer-derived A2780 cells. In addition, the knockdown of SPRY2 inhibited the cell proliferation [57]. In another study, Zhao et al. showed that GAS5 was downregulated in ovarian cancer tissues and cell lines and associated with larger tumor size and advanced FIGO stage (III-IV). Overexpression of GAS5 significantly repressed cell growth, proliferation, and activates apoptosis, whereas its silencing produced the opposite effects. In addition, it was found that miR-196-5p expression was negatively associated with GAS5 and that miR-196-5p negatively regulates the expression of HOXA5 via direct binding to its 3’-UTR [58]. These data suggested that GAS5 promoted ovarian cancer progression by modulating the miR-196-5p/HOXA5 axis making it an attractive target for therapy.

### 3.6. HOXD Cluster Antisense RNA 1 (HOXD-AS1)

LncRNA HOXD cluster antisense RNA 1 (HOXD-AS1) is transcribed from the HOXD gene cluster located at human chromosome 1q31.2 [59]. Recent studies have shown that HOXD-AS1 is overexpressed in gynecological cancers such as uterine, ovarian, and bladder [54,55,56,60,61,62], promoting cell growth, migration, EMT, and invasion [60]. Zhang et al. reported that expression of HOXD-AS1 was upregulated in cancer tumors and cell lines and it was correlated with advanced FIGO stage, lymph node metastasis, and poor overall survival in ovarian cancer patients. Furthermore, the tumor suppressor miR-133a-3p was downregulated in ovarian cancer tissues and cell lines and negatively correlated with HOXD-AS1 expression. The luciferase assays confirmed that HOXD-AS1 sponges to miR-133a-3p. In addition, HOXD-AS1 promoted cell proliferation, invasion, and EMT through miR-133a-3p and activation of Wnt/β-catenin signaling [61].

A similar study confirms the upregulation of HOXD-AS1 in ovarian cancer tissues and cell lines and predicted a poor prognosis. Similarly, inhibition of HOXD-AS1 reduced cell migration, invasion, and EMT by preventing HOXD-AS1 binding to miR-186-5p, and resulted in downregulation of PIK3R3. These data suggested that HOXD-AS1 influences ovarian cancer progression by modulating the miR-186-5p/PIK3R3 axis [63].

### 3.7. Lung Cancer-Associated Transcript 1 (LUCAT1)

Lung cancer-associated transcript 1 (LUCAT1) was first described in non-small lung cancer cells [64]. In addition, its oncogenic role has also been described in glioma, esophageal squamous cell carcinoma, and clear cell renal cell carcinoma [65,66]. Interestingly, two reports highlight the role of LUCAT1 in ovarian cancer. Yu et al. demonstrated that LUCAT1 expression increased significantly in ovarian cancer tissues and this was positively associated with metastasis and clinical stage. The repression of LUCAT1 inhibited proliferation, invasion, and promoted apoptosis through the LUCAT1/miR-612/HOXA13 axis. Mechanistically, LUCAT1 sponges to miR-612, which in turn binds and represses the HOXA13 transcript [67]. In other study, Liu et al. also demonstrated that LUCAT1 was upregulated in ovarian cancer cells. The knockdown of LUCAT1 significantly repressed proliferation and colony formation and activates apoptosis. In addition, a highly conserved binding site of miR-199a-5p was found in the 3’-UTR of lncRNA LUCAT1. These data suggested that LUCAT1 promotes ovarian cancer progression by modulating the miR-199a-5p [68].

### 3.8. LncRNA Colon Cancer-Associated Transcript 1 (CCAT1)

The lncRNAs colon cancer-associated transcript 1 and 2 (CCAT1 and CCAT2) are located within the 8q.24.21 chromosome around the MYC proto-oncogene region [69]. The high expression of CCAT1 and CCAT2 was significantly associated with recurrence-free survival and overall survival in colon cancer, gastric cancer, hepatocellular carcinoma, breast cancer, and ovarian cancers [70,71,72]. CCAT1 and CCAT2 promoted the metastasis and proliferation of different cancers through the interaction with c-Myc, sprouty homolog 4 (SPRY4)14, and several microRNAs [71,73]. For example, Lai et al. demonstrated that CCAT1 was upregulated in ovarian tumors, compared with normal adjacent tissues. Moreover, a high expression correlated with poor prognosis, based on tumor size and lymph node metastasis. The knockdown of CCAT1 inhibited cell proliferation and migration in ovarian cancer cells. In addition, it was found that miR-1290 expression was negatively associated with CCAT1. [74]. Similarly, CCAT2 expression was upregulated in tissues and cell lines of ovarian cancer. The knockdown of CCAT2 suppressed cell proliferation and promoted apoptosis in cancer cells. In addition, the luciferase assay demonstrated that CCAT2 sponges to miR-424. Interestingly, the miR-424 inhibitor rescued the tumorigenesis inhibition induced by CCAT2 knockdown [75].

### 3.9. Other LncRNAs/MicroRNAs Pairs Involved in Ovarian Cancer Progression

Several studies report on the role of less-characterized individual lncRNAs and its target microRNAs in the development and progression of ovarian cancer. For instance, Liu et al. demonstrated that the lncRNA-regulator of reprogramming (ROR) showed high expression in ovarian cancer biopsies relative to normal ovarian tissues. ROR regulated cell migration and invasion by inducing EMT via suppression of miR-145 and release of its posttranscriptional target filamin B protein (FLNB), an actin-binding protein involved in cytoskeleton functions [76]. Another study showed that imprinted lncRNA H19 was upregulated, whereas miR-370-3p was suppressed after stimulation of cancer cells with transforming growth factor beta (TGFβ), a well-known EMT activator. Knockdown of H19 or miR-370-3p overexpression impaired TGFβ-induced EMT, whereas H19 upregulation or a miR-370-3p suppression resulted in TGFβ induction of EMT program. Functionally, it was found that H19 targets miR-370-3p acting as a bona fide ceRNA [77].

The RNA prostate cancer-associated transcript 1 (PCAT1, also known as PCAT8 and CARLo3) is a lncRNA of ~1900 nt in length mapped at chromosome 8q24 and 725 kb upstream of the c-Myc oncogene in a genomic region containing the prostate cancer susceptibility SNP loci [78]. LncRNA-PCAT1 is oncogenic and promotes cell proliferation, migration, and invasion by regulating Wnt/β-catenin-signaling in diverse types of cancers, including esophageal squamous cell carcinoma, hepatocellular cancer, non-small cell lung cancer, glioblastoma, breast cancer, colorectal cancer, and gastric cancer. Min et al. showed that PCAT1 was overexpressed in ovarian cancer tissues. Remarkably, PCAT1 silencing in A2780 and SKOV cell lines leads to impairment of cell proliferation, migration, and invasion, while apoptosis was activated, which was accompanied by increased expression of miR-124-3p and downregulation of cyclin D1, CDK6, p53, Bax, cleaved caspase- 3, metallopeptidases, and vimentin proteins involved in migration and cell death. Moreover, it was shown that PCAT1 knockdown produced a significant decrease in Wnt3a, β- catenin, and phosphorylation of protein kinase B [79].

Previously, it was reported that a novel lncRNA named as AB209371 was overexpressed in metastatic liver cancer promoting the EMT [80]. In a related study, Zheng et al. showed that lncRNA-AB209371 overexpression impaired the expression of miR-203 which in turn derepressed its target survivin resulting in augmented proliferation of ovarian cancer cells. Congruently, the ectopic expression of miR-203 produced the opposite results and attenuated the effects of AB209371 overexpression [81]. Moreover, it was recently demonstrated that lncRNA 460 (LINC00460) was upregulated in tissues and cell lines with epithelial ovarian cancer. Deletion of LINC00460 inhibited cell proliferation, migration, and invasion, as well as induced apoptosis through upregulation of miR-338-3p [82]. Likewise, the lncRNA SNHG12 (small nucleolar RNA host gene 12) was found as overexpressed in several types of cancer including ovarian cancer [83].

Overexpression of SNHG12 promoted the proliferation and migration of A2780 and HO8910 cells, through the negative regulation of miR-129, promoting overexpression of its target mRNA SOX-4 [84]. Sun et al. showed the overexpression of SNHG12 in ovarian cancer tissues and cell lines. Interestingly, patients with metastatic ovarian cancer or those with stage III-IV had a higher level of SNHG12 compared with nonmetastatic or stage I-II patients. Similarly, the lncRNA GIHCG was upregulated in ovarian cancer tissues, whereas microRNA-429 was repressed. GIHCG overexpression in HEY cells promoted cell cycle and colony-forming ability. Interestingly, overexpression of miR-429 inhibited cell cycle and colony-forming ability in ovarian cancer cell lines [85]. Finally, it has been shown that nuclear enriched abundant transcript 1 (NEAT1) was overexpressed in ovarian cancer tissues and cell lines. Liu et al. demonstrated that silencing of NAEAT1 inhibited metastasis and expression of Rho-associated coiled-coil-containing protein kinase 1 (ROCK1) by functioning as a ceRNA for miR-382-3p [86].

On the other hand, ADAMTS9-AS2 lncRNA was significantly reduced in ovarian cancer tissues and cell lines and it was correlated with advanced FIGO stage, lymph node metastasis, and poor overall survival of ovarian cancer patients. Furthermore, Wang A et al. demonstrated that repression of ADAMTS9-AS2 inhibited the cell proliferation, invasion, and EMT processes in vitro and restricted tumor growth in vivo via the miR-182-5p/FOXF2 axis [87].

## 4. LncRNAs and MicroRNAs Regulating Angiogenesis in Ovarian Cancer

Tumoral angiogenesis has an important functional role in tumor growth and metastasis and predicts a worst prognosis for patients [88]. Recent evidences demonstrated that lncRNAs and miRNAs regulate tumor cell angiogenesis in ovarian cancer (Figures 2 and 3). Several studies gave insights on the role of proinflammatory chemokines in the development of vascular blood supply at the site of inflammation in cancer tissues. Song et al. showed that chemokine CXCL12 was upregulated, while miR-137 was downregulated in ovarian cancer tissues, and associated with poor prognosis. Functionally, miR-137 directly binds to CXCL12 to regulate cell invasion and angiogenesis in ovarian cancer [89].

The differentiation antagonizing non-protein coding RNA (DANCR) is an oncogenic lncRNA overexpressed in various cancers. Results demonstrated that miR-145, a tumor suppressor involved in tumor angiogenesis, is a direct target of DANCR promoting the derepression of vascular endothelial growth factor (VEGF) in ovarian cancer [76]. On the other hand, anisomycin an anti-protozoal antibiotic extracted from Streptomyces griseolus may have the potential to inhibit the cancer stem cells. The study demonstrated that there was a positive correlation between tumor angiogenesis and the number of CD44+/CD133+ ovarian cancer stem cells. The anisomycin treatment significantly inhibited the expression of Notch pathway components and vascular endothelial cell markers. LncRNA-Meg3 showed high expression in ovarian cancer cells, which binds to endogenous miRNA-421 and blocks its inhibitory effect on downstream target genes including platelet derived growth factor receptor α (PDGFRA). However, after cancer stem cells were treated with anisomycin, the expression of lncRNA-Meg3 was inhibited, resulting in the attenuation of its sponge effect on miRNA-421 and tumor angiogenesis [90].

## 5. Role of the LncRNAs/MicroRNAs in Therapy Resistance of Ovarian Cancer

The remission of women with ovarian cancer is highly challenging due to the late diagnosis and resistance to anticancer drugs. Therefore, it is critical to clarify drug resistance mechanisms in order to increase the prognosis of ovarian cancer patients. Recent reports have evidenced the participation of dysregulated lncRNAs/microRNAs interactions in therapy resistance. By transcriptomics profiling of cisplatin-sensitive and cisplatin-resistant ovarian cancer samples, Li et al. confirmed the link between dysregulation of lncRNAs and drug resistance. Notably, they found the overexpression of urothelial carcinoma associated 1 (UCA1), a lncRNA that has been previously reported as upregulated in different types of cancers and with a role in drug resistance. Loss- and gain-of-function experiments confirmed that UCA1 expression was required for cisplatin resistance in ovarian cell lines. Functional experiments indicated that UCA1 drives cisplatin resistance by sponging miR-143, which results in a reduced expression of FOS-related antigen 2 (FRA-2/FOSL2), one of the target genes of miR-143. FOSL2 is a leucine zipper DNA-binding of the AP-1 transcription factor family which has been previously involved in progression and chemoresistance in other types of cancer. One important finding of this work is the detection of overexpressed UCA1 in extracellular vesicles namely exosomes, which opens the possibility to use this lncRNA as a circulating biomarker for cisplatin-resistance in women with ovarian cancer [91].

Another group of investigation also identified linc00161 among the set of overexpressed lncRNAs in cisplatin-resistant ovarian tumors. By performing similar experiments, they showed that Linc00161 stimulates the drug resistance of ovarian SKOV3 cells by sponging miR-128, which allows the expression of its target gene, MAPK1. These data suggest that the MAPK signaling cascade might have a role in drug resistance [92]. Importantly, these two studies evidenced that the ectopic expression of UCA1 or Linc00161 suppressed the tumorigenicity of ovarian cancer and reduced the cytotoxic effects of cisplatin in a tumor xenograft murine model.

In a recent paper, Zhang et al. evidenced that the expression of HOTAIR was increased in cisplatin-resistant SKOV3 and A2780 cell lines, while the amount of miR-138-5p whose sequence is partially complementary to HOTAIR was decreased. The forced expression of miR-138-5p increased cisplatin sensitivity by targeting EZH2 and SIRT1 genes involved in the regulation of cisplatin resistance of ovarian cancer. Knockdown of HOTAIR produced similar effects that were reversed by silencing of miR-138-5p [93].

In addition to these experiment strategies, a bioinformatics approach can be an efficient tool to predict the lncRNA-miRNA-mRNA regulatory network related to cisplatin resistance in ovarian cancer cells. By screening public data sets, Zhao et al. identified 485 differentially expressed genes, 85 lncRNAs, and five miRNAs, some of them previously related to drug resistance in cancer. Notably, four of the five miRNAs (miR-29a-3p, miR-128-3p, miR-7-5p, and miR-140-3p) target several modulated genes including JUN, BCL2, HLA-F, and TRIM34 that were identified as the most important genes in the protein-protein interaction network constructed by STRING. Additionally, DIANA-LncBase V2 predicted that four lncRNAs (XLOC_003240, MIAT, XLOC_003204, and XLOC_010743) are potentially able to sponge these four miRNAs. Interestingly, several members of this predicted lncRNA-miRNA-mRNA regulatory network have been previously related to chemoresistance in human cancers, which support the relevance of this bioinformatics strategy to identify new molecules that are worth being evaluated in cisplatin resistant ovarian cancer cells. However, experimental assays in cell lines and human tissues are needed to confirm these findings [94].

In the past few years, various dietary supplements have been shown to have an antitumoral activity and produce chemo-sensitization of cancer cells. In an interesting paper, Zhang et al. investigated how curcumin affects extracellular vesicles to suppress cisplatin resistance. Extracellular vesicles such as exosomes can transport proteins, mRNA, and non-coding RNA from a donor cell cytoplasm to another cell, with the possibility of modifying its gene expression and phenotype. From lncRNA expression profiling of extracellular vesicles purified from ovarian cancer A2780 cells cultured with and without the curcumin, they showed changes in the expression of a large number of lncRNAs, being MEG3 one of the most upregulated due to demethylation in its promoter region. Both curcumin treatment and MEG3 ectopic expression resulted in decreased levels of miR-214 that has two confirmed binding sites in MEG3, cells, and vesicles. Inhibition and overexpression of miR-214 affected the ability of vesicles to regulate apoptosis and cell survival in the presence of cisplatin. Altogether, these results evidenced that curcumin, can revert this ability by upregulating MEG3 to sponge miR-214 previously involved in the PTEN/AKT pathway and the p53/Nanog axis [95].

Shi et al. reported a lncRNAs genomic profiling study in paclitaxel resistant ovarian cancer cells. Notably, they evidenced that lncRNA-LINC01118 is overexpressed in paclitaxel, cisplatin, and taxol resistant ovarian cells. Its overexpression in patient biopsies also showed its participation in cancer development. Up- and downregulation of LINC01118 in sensitive and resistant SKOV3 cells, respectively, was able to modulate the paclitaxel sensitivity. Functional assays confirmed that LINC01118 acts as an oncogene, since it negatively regulated apoptosis and promoted invasion. LINC01118 modulated paclitaxel resistance by sponging miR-134 that targets the ATP Binding Cassette C1 (ABCC1), an ATP-dependent efflux pump that drives drug resistance in a large number of cancers, allowing the efflux of cisplatin and cell survival [96].

Two different groups reported a higher amount of UCA1, a lncRNA previously involved in cisplatin resistance, in Paclitaxel resistant cells, suggesting that this lncRNA may have a central role in drug resistance in ovarian cancer [91,92,97,98]. Congruently, UCA1 silencing reduced the ability of paclitaxel resistant SKOV3 and HeyA-8 cells to grow in the presence of paclitaxel by promoting apoptosis and inhibiting proliferation, migration, and invasion. Additionally, there was an upregulation of miR-219 and miR-654-5p that have complementary binding sequences in UCA1, which suggest that UCA1 knockdown enhanced paclitaxel sensitivity by suppressing miR-129 and miR-654-5p. Interestingly, the ABCB1 was directly targeted by miR-129, while miR-654-5p targeted Salt inducible kinase 2 (SIK2) which is known to promote metabolism and metastasis in ovarian cancer. Functional experiments of silencing or overexpression of UCA1 or miRNAs, confirmed that UCA1 positively regulates ABCB1 and SIK2 expression by sponging miR-129 and miR-654-5p, respectively. They also suggested that UCA1 knockdown could sensitize paclitaxel-resistance and improve patient response to chemotherapy [97,98].

As previously described in other types of cancer, the expression of the lncRNA taurine-upregulated gene 1 (TUG1) was elevated in ovarian cancer biopsies and A2780 and SK-OV-3 cell, as well as in Paclitaxel resistant cells. TUG1 overexpression was confirmed as a marker of an unfavorable prognosis. As expected, TUG1 suppression increased PTX sensitivity in vitro by promoting autophagy. Interestingly, the expression of TUG1 and miR-29b-3p, one of its potential targets, exhibited an inverse correlation in ovarian cancer tissues and drug resistant cells, suggesting that TUG1 acts as a miR-29b-3p sponge. In this study, the establishment of a xenograft tumor model in nude mice using paclitaxel resistant A2780 cells with or without TUG1 silencing, confirmed that TUG1 promoted cancer progression and paclitaxel resistance in vivo by inhibiting miR-29b-3p [99].

LncRNA zinc finger antisense 1 (ZFAS1) that has been shown to participate in the development and progression of several cancers was also found overexpressed in ovarian cancer tissues and associated with poor prognosis, by enhancing cell proliferation, migration, and invasion. Moreover, knockdown of ZFAS1 improved cisplatin and paclitaxel-sensitivity in ovarian cancer cells, indicating an additional role in chemoresistance. Consistently, all these events were reduced when miR-150-5p, a target of ZFAS1, was overexpressed, and the expression of the transcription factor Sp1 (Specificity protein 1), a target of miR-150-5p was rescued. Therefore, the ZFAS1/miR-150-5p/Sp1 axis is responsible, at least in part, for the regulation of cell malignancy and cisplatin and paclitaxel-resistance in ovarian cancer cells in vitro [100].

The fluoropyrimidine 5-fluorouracil (5-FU) that is tested as an anti-ovarian cancer drug in clinical protocols, inhibits the thymidylate synthase (TS) and interferes with DNA and RNA synthesis. In a recent global transcriptomics study, Li et al. found several lncRNAs modulated in 5-FU resistant SKOV3 (SKOV3/5-FU) cells [95,101]. Among them, the antisense lncRNA named Thymopoietin (TMPO)-antisense RNA1 (TMPO-AS1) that is known to promote cell proliferation and decrease apoptosis was overexpressed in 5-FU-resistant cells and also in ovarian tumor tissues. Congruently, EMT invasion, migration, and drug resistance were inhibited when TMPO-AS1 was silenced. TMPO-AS1 acts as a sponge lnRNA that sequestered miR-200c and induced the overexpression of its gene target, the transmembrane protein with epidermal growth factor and two follistatin motifs 2 (TMEFF2), which promoted drug resistance and metastasis in SKOV3/5-FU cells. The PI3K/Akt signaling pathway was also found activated, however the relationships between TMEFF2 and the PI3K/Akt signaling pathway and cancer progression remain unclear [101].

## 6. Regulation of Warburg Effect by LncRNAs in Ovarian Cancer

The Warburg effect refers to the metabolic adaptations that develop the cancer cells in which, regardless of the O2 cellular levels, they tend to utilize high quantities of glucose to the glycolytic pathway for ATP production followed by lactate generation, rather than the use of mitochondrial TCA cycle and the respiration chain [102]. This metabolic rewiring stimulates cell proliferation and growth, promotes metastasis, increases cancer cell survival, and long-term tumor maintenance. Two recent studies showed the functional effects of lncRNAs in the Warburg effect in ovarian cancer cells. In the first one, Zheng et al. studied the effects of the ginsenoside 20(S)-Rg3, a saponin obtained from red ginseng which displays broad anti-tumor effects. After high-throughput RNA sequencing of 20(S)-Rg3-treated and non-treated control SKOV3 ovarian cancer cells, 18 significantly increased lncRNAs and 67 downregulated lncRNAs were identified [103]. Of these, lncRNA H19 was one of the most severely suppressed and further characterized. Functional studies showed that the specific knockdown of H19 resulted in reduced glucose consumption and lactate production, as well as decreased PKM2 levels. In contrast, lncRNA H19 overexpression in 20(S)-Rg3-treated ovarian cancer cells produce the opposite effects. Mechanistically, it was found that H19 binds to miR-324-5p which in turns targets the 3’-UTR of PKM2 transcript. Finally, it was demonstrated that miR-324-5p overexpression efficiently blocked the cell proliferation in the in vitro and in vivo models [103].

In another study, the functions of lncRNA LINC00504 were investigated in ovarian cancer. LINC00504 expression was significantly increased in ovarian cancer cell lines and tumor tissues [104]. The specific silencing of LINC00504 resulted in the inhibition of cell proliferation, enhanced apoptosis, reduced expression of PKM2, HK2, and PDK1 genes, and aberrant aerobic glycolysis. Functionally, the lncRNA LINC00504 sequestered and downregulated miR-1244 levels. Contrarily, the forced inhibition of miR-1244 diminished the effects of LINC00504 in ovarian cancer cells. 

Altogether these studies showed a relevant role for lncRNAs/microRNAs in cell migration, angiogenesis, Warburg effect, and therapy resistance (Figure 3).

## 7. LncRNAs/MicroRNAs Coregulation Networks

It is well known that lncRNAs and microRNAs play key functional roles by interacting with other molecules forming complex interactions-networks, finely coordinated and regulated, to jointly modulate the biological functions within the cells. These ncRNAs and protein-coding RNAs species can interconnect with each other through different mechanisms: (i) MiRNAs bind to mRNAs or lncRNAs in a sequence-specific manner activating the RNA interference mechanism resulting in degradation or inhibition of translation of their RNAs targets; (ii) lncRNAs can interact with mRNAs and participate in its post-transcriptional regulation such as processing, transport, translation, and decay; and (iii) lncRNAs contain multiple binding sites for different miRNAs, functioning as a sponge or ceRNA. In this case, when the miRNAs molecules are kidnapped by lncRNAs their multiple cognate mRNA targets are derepressed and available for translation impacting in this way the cancer phenotypes [105]. Both, aberrant expression and dysfunction of individual lncRNAs and microRNAs have been associated with different hallmarks of cancer [106]. However, the joint participation of these genetic regulators is still poorly explored.

Until recently, most of the studies were limited to describing the clinical and functional relevance of specific lncRNAs and microRNAs pairs or a small set of them. Guo and cols [107] analyzed and integrated lncRNAs and microRNAs genomic profiles reported in TCGA and reported for the first time the validation of a signature, formed by four ncRNAs (two lncRNAs and two miRNAs), as a predictor of survival in a cohort of ovarian cancer patients without BRCA1/2 mutations. They reported that lncRNAs (LINC01234 and CCDC144NL-AS1) and miRNAs (miR-637 and miR-129-5p) can serve together as a marker of risk stratification (high and low risk) of patients with wild-type BRCA1/2, a highly-aggressive cancer subtype with low prognosis. Shortly after, another group used a similar approach and mined the TGCA databases for proteins, lncRNAs, and microRNAs associated with the prognosis of patients with ovarian serous carcinoma. They reported the validation of a signature made up of protein-coding mRNA transcripts (VAT1L, CALR), lncRNAs (LINC01456, RP11-484L8.1), and miRNAs (miR-196A1 and miR-148A) to potential risk stratification of patients with ovarian serous carcinoma, a type of tumor that originates from ovary cells located in the serous epithelial layer [108].

The high-grade serous ovarian carcinoma (HGSC) subtype frequently involves a fatal peritoneal tumor spread and presents the lowest survival rates [109]. Lou et al. obtained the expression profiles of lncRNAs and microRNAs by the microarrays analysis specific for this condition and healthy fallopian tube tissues. The differences in expression of a set of lncRNAs and mRNAs were confirmed to be altered between HGSC cancer tissues. Then, a set of five differentially expressed lncRNAs (GTSE1-AS1, FAS-AS1, AK130076, RP11-199F11.2 and AC093818.1) were selected to further explore their expression in relation with the clinical parameters of patients [110]. Results showed a significant association between lncRNAs expression and clinicopathological characteristics, suggesting the possible participation in the development of ovarian carcinogenesis and tumor progression. On the other hand, HGSC has been subclassified into miliary and non-miliary, depending on peritoneal mode dispersion. To understand the molecular mechanisms of peritoneal tumor spread and identify small RNAs associated with the two types of peritoneal tumor spread in HGSC, Bachmayr-Heyda et al. performed a small RNA sequencing (sRNA-seq) and constructed a ceRNA network [111]. For this, fresh tumor tissue cells from ovary, peritoneum, and ascites were isolated and analyzed and a group of 13 small RNAs was selected as a spread predictor in HGSC. The gene signature consists of three microRNAs higher expressed in miliary (miR-760, miR-1003, miR-2508), and eight microRNAs (miR-1254, miR-804, miR-1628, miR-1927, miR-2353, miR-2916, miR-3475, miR-3784) and two piRNAs (piRno_hsa_009295 and piRn9_chr19_12817301) higher expressed in non-miliary condition. The identified sRNA predictors can be used to classify ovarian cancer patients and assist in establishing a better initial treatment.

To investigate the molecular mechanisms that trigger ovarian cancer as well as the prognostic value of interaction networks in this pathology, Wang et al. constructed a ceRNA network [112]. Gene expression information were obtained from RNA-sequencing profiles stored in TCGA and GEO databases. Differentially expressed genes between recurrent and non-recurrent tumor samples were identified and analyzed to identify prognostic biomarkers. Three lncRNAs (such as WT1-AS), six miRNAs (such as miR-375, miR-141, and miR-27b), and 58 mRNAs (including TP53 and RBPMS) were significantly associated with prognosis, and can be experimentally studied as prognostic predictors of recurrent ovarian cancer. Moreover, WT1-AS-miR-375-RBPMS and WT1-AS-miR-27b-TP53 axes were identified as probable routes of recurrence appearance in ovarian cancer patients. In other study, Li et al. established gene interaction networks from lncRNAs, microRNAs, and mRNAs expression profiles using data from GEO databases [113]. They discovered that WDFY3-AS2 lncRNA is a ceRNA of miR-18a, their interaction promotes the overexpression of its RORA target gene, increasing apoptosis, and inhibiting cell proliferation, invasion, and EMT. These data suggest that WDFY3-AS2 overexpression may be a strategy to reduce the development of ovarian tumors.

To identify molecules that serve as biomarkers and reveal new mechanisms that govern this condition, Li et al. performed an analysis of ceRNA networks from epithelial samples of ovarian cancer [114]. They collected the mRNA, microRNAs, and lncRNAs expression data from the GEO database obtained from microarrays analyses in normal and cancer tissues samples. The results obtained show that at least 14 differentially expressed miRNAs could potentially bind to 124 differentially expressed lncRNAs and 92 differentially expressed mRNAs. The network predicts 164 nodes (interactions), the components identify with a higher number of nodes that potentially may be utilized as diagnosis and prognosis biomarkers in ovarian cancer.

Recently, Zhou et al. performed an integrative network with 20,424 mRNAs, 10,412 lncRNAs, and 742 miRNAs differentially expressed in 407 ovarian cancer patients from TCGA [115]. In summary, 20,424 mRNA-lncRNA-miRNA interactions were predicted within the network, and 7311 cliques (a triplet consisting of one mRNA, one miRNA, and one lncRNA interrelated physically and functionally) were identified. Thereafter, cliques were analyzed in correlation with the survival and stage of patients, this correlation was positive for nine cliques suggesting that they play a role in ovarian carcinogenetic mechanisms. To explore the biological meaning of cliques selected, a functional analysis was performed, showing that components can contribute to the initiation and progression of ovarian cancer through regulating cell cycle, growth, proliferation, migration, multidrug resistance, and angiogenesis.

## 8. Discussion

The identification of complex coregulation networks can provide a broader view of the role of RNAs regulators in the modulation of gene expression in complex diseases, such as cancer. When lncRNAs-miRNAs-mRNAs interaction-networks are altered because the expression and availability of one or more network components is modified, the cell phenotype can change and trigger or contribute to the onset of several pathologies. In ovarian cancer, the abnormal regulation of ncRNAs drives cellular transformation and tumor progression. Tumor cells form different interaction networks, compared to normal cells, so the identification of players and/or specific interaction-networks that occur both in normal and pathological conditions is of great interest. The identification of aberrant lncRNAs-miRNAs-mRNAs interactions in cancer can serve as an important predictive tool for diagnosis, prognosis, treatment, and therapy response (Figure 4). The recent studies described above revealed the existence of various lncRNAs/microRNAs coregulation networks responsible for drug resistance in ovarian cancer cells. The fact that various lncRNA/miRNA pairs can promote different molecular mechanisms to overcome the effects of one drug, i.e., cisplatin or paclitaxel that are the most commonly used, indicates that drug resistance in ovarian cancer is a very intricate phenomenon. On the other hand, distinct lncRNA/miRNA pairs can also activate the same molecular mechanism, such as the ABC proteins that can mediate the active efflux of a variety of drugs, which adds even more complexity to this problem, seriously affecting patient survival.

Importantly, the modulation of identified lncRNAs may represent a valuable strategy for ovarian cancer treatment. The presence of dysregulated lncRNAs in EV circulating in serum also suggests that they may serve as biomarkers for early drug resistance prediction without invasion procedures, allowing a rapid switch in therapeutic strategy to improve patient survival. However, these studies have some limitations. Even if dysregulation of several lncRNAs is confirmed in ovarian tumors, the elucidation of the lncRNAs/miRNAs/mRNAs axis involved in chemoresistance is mostly demonstrated in vitro, and there are no validation studies in samples from drug resistant patients. Moreover, most studies only focused on one drug, therefore it would be of interest to verify whether the identified lncRNAs/microRNAs coregulation networks also control resistance to other drugs. In conclusion, all the individual lncRNAs/microRNAs pairs and interaction networks described here may represent potential therapeutic targets for clinical intervention.

## Figures and Tables

**Figure 1 cells-10-01407-f001:**
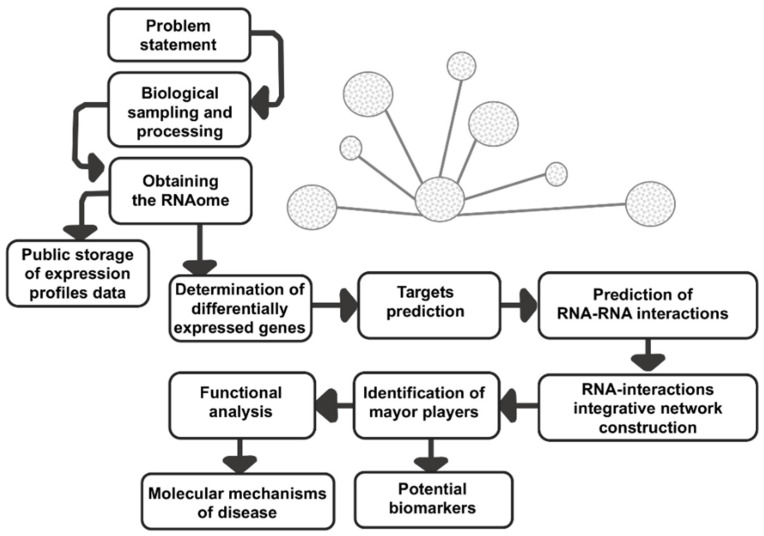
Flowchart of a typical ncRNAs-interactions network analysis. RNAome can be obtained from RNA-seq or microarrays analysis, and from databases such as The Cancer Genome Atlas (TCGA) or Gene Expression Omnibus (GEO). The prediction of gene targets is routinely performed with programs such as lncRNASNP, miRsponge, NONCODE, miRmine, starBase, and PeTMbase, etc. The prediction of lncRNAs-miRNAs-mRNAs interactions networks can be built using computational programs, typically Cytoscape. Subsequently, the gene ontology analysis can be performed in order to identify genetic pathways and altered cellular functions. Finally, the experimental validation is a necessary step to obtain a more complete portrait of ncRNAs interplay resulting in specific cancer phenotypes.

**Figure 2 cells-10-01407-f002:**
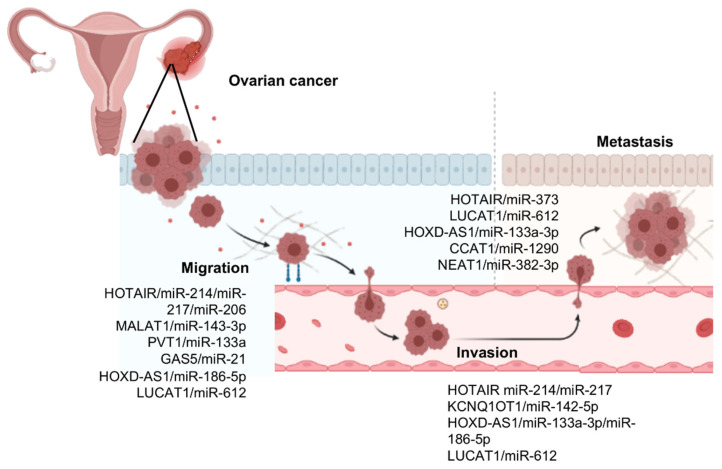
LncRNAs/microRNAs pairs deregulated in ovarian cancer. The image indicates the relevant lncRNAs/microRNAs interactions reported on the regulation of cancer cell migration, invasion, and metastasis.

**Figure 3 cells-10-01407-f003:**
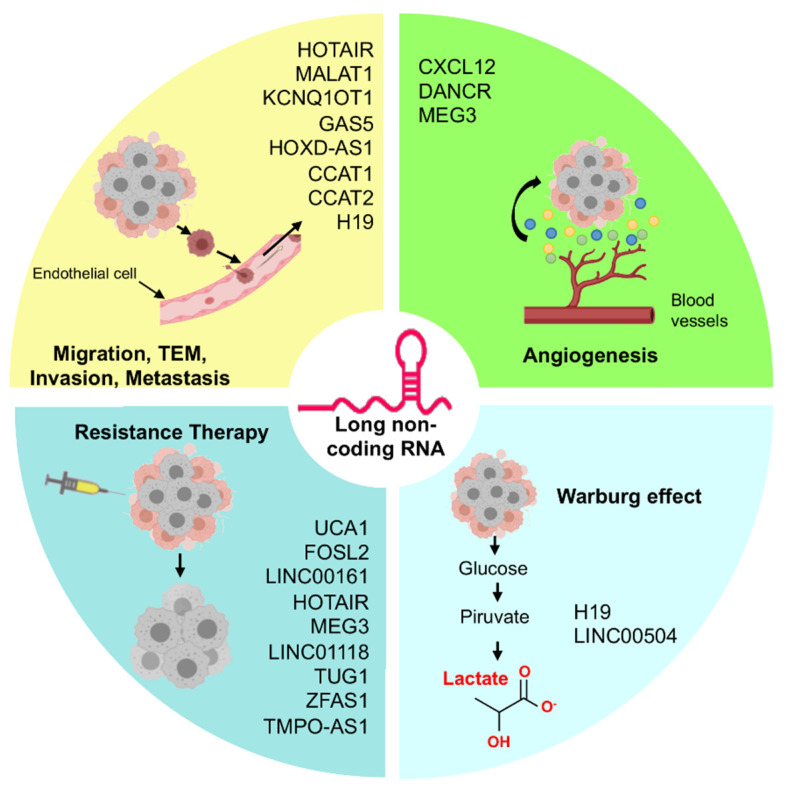
LncRNAs deregulated in ovarian cancer. Image shows the main lncRNAs found as deregulated in ovarian cancer cells and tissues affecting the diverse cancer hallmarks.

**Figure 4 cells-10-01407-f004:**
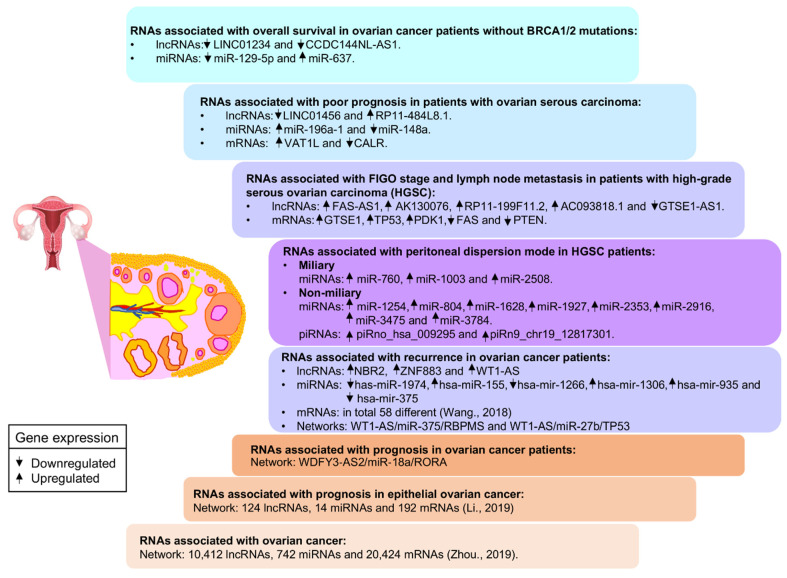
Use of coregulatory networks for the identification of ncRNAs-based clinical predictors. The analysis of the coregulatory networks allows identifying specific expression profiles of different RNA species for a given condition, and serving as clinical predictors, such as diagnosis, prognosis, recurrence, and survival of patients with cancer.

## Data Availability

Not applicable.

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
