# Peer review of "Deciphering the Long Non-Coding RNAs and MicroRNAs Coregulation Networks in Ovarian Cancer Development: An Overview"

_cells, 2021, doi:10.3390/cells10061407_

Round 1

Reviewer 1 Report

The review entitled “Deciphering the Long Non-Coding RNAs and MicroRNAs Coregulation Networks in Ovarian Cancer Development: An Overview, by César López-Camarillo is very well written and will certainly be very helpful to the researchers of the filed.

For the betterment of the review I will suggest –

  1. Please elaborate the section 2 “The expanding repertoire of regulatory non-coding RNAs: lncRNAs and microRNAs”, page 2 lines 81 onwards.

Here the author has described about the nuclear lncRNAs please add about the cytosolic lncRNA as well, and describe what are the factors deciding nuclear vs cytosolic lncRNA.

  1. Please add justification why you want to keep and discuss the section 8 “LncRNAs and microRNAs in polycystic ovarian syndrome” in the review.

Author Response

Reply to Reviewer 1

The review entitled “Deciphering the Long Non-Coding RNAs and MicroRNAs Coregulation Networks in Ovarian Cancer Development: An Overview, by César López-Camarillo is very well written and will certainly be very helpful to the researchers of the filed.

Reply: We acknowledge to editors and reviewer for the opportunity to revise the manuscript. Your critical suggestions that we have fully replied greatly increase the quality of our study. All amendments have been marked in yellow color in the revised text for your easy reference and reading. We have carefully reviewed the manuscript according the referee suggestions and provide a point-by-point response.

For the betterment of the review I will suggest –

  1. Please elaborate the section 2 “The expanding repertoire of regulatory non-coding RNAs: lncRNAs and microRNAs”, page 2 lines 81 onwards.

Reply. We appreciate the reviewer’s comments. however, we think that section 2 "The expanding repertoire of regulatory non-coding RNAs: lncRNAs and microRNAs", should start on page 2, line 70 onwards, as this first paragraph describe the biology of lncRNAs as the subtitle establish.

Here the author has described about the nuclear lncRNAs please add about the cytosolic lncRNA as well, and describe what are the factors deciding nuclear vs cytosolic lncRNA.

Reply: Thanks, you very much to reviewer for pointed-out this important issue. We have included new information about the cytosolic lncRNAs which describes their functions and localization.

  1. Please add justification why you want to keep and discuss the section 8 “LncRNAs and microRNAs in polycystic ovarian syndrome” in the review.

Reply. Thanks, you very much for the comments. As the reviewer 2 also has concerns about the pertinence of this section, we have decided to remove it as we consider that it is better to focus the review only on ovarian cancer.

Reviewer 2 Report

In this manuscript, Lopez-Camarillo et al. explored the existing literature regarding coregulation networks between lncRNAs and microRNAs in ovarian cancer. Generally speaking, this work has good potential if the following points are addressed:

  1. The manuscript needs additional editing. Minor grammatical errors that need to be revised. For example, see lines 70, 104, 144-147, but there are others. Figure 1 “identification of mayor players.”
  2. The manuscript is too long. It would benefit from a more concise description for each LncRNA section.
  3. Even though the authors provide an adequate explanation for reviewing LncRNAs and microRNAs in ovarian cancer, the rationale, and relevance of reviewing LncRNA/microRNA pairs in the context of ovarian cancer is not that strong. An enhanced introduction to this topic targeting non-specialists will strongly help the reader.
  4. The authors stated that they are reviewing LncRNA/microRNAs, which have at least two papers published, which seems a very arbitrary selection method. Are there other key LcnRNA/microRNA pairs worth mentioning?
  5. The review reads like a list of LcnRNA/microRNA pairs and a summary of published manuscripts. I would prefer to see a more concise summary of published work and an expanded discussion on the relevance and potential of targeting these interactions.
  6. I would remove the LcnRNA/microRNA in polycystic ovarian syndrome as the manuscript can benefit from focusing solely on ovarian cancer.

Author Response

Reply to Reviewer 2

In this manuscript, Lopez-Camarillo et al. explored the existing literature regarding coregulation networks between lncRNAs and microRNAs in ovarian cancer. Generally speaking, this work has good potential if the following points are addressed:

Reply: We acknowledge to editors and reviewer for the opportunity to revise the manuscript. Your critical suggestions that we have fully replied greatly increase the quality of our study. All amendments have been marked in yellow color in the revised text for your easy reference and reading. We have carefully reviewed the manuscript according the referee suggestions and provide a point-by-point response.

  1. The manuscript needs additional editing. Minor grammatical errors that need to be revised. For example, see lines 70, 104, 144-147, but there are others. Figure 1 “identification of mayor players”.

Reply. Thank you very much for your comment, we have revised and corrected the grammatical errors throughout the manuscript.

  1. The manuscript is too long. It would benefit from a more concise description for each LncRNA section.

Reply: As reviewer suggested, we shortened the manuscript from 26-to-20 pages. Several long paragraphs were shortened making more concise and succinct the description of data.

  1. Even though the authors provide an adequate explanation for reviewing LncRNAs and microRNAs in ovarian cancer, the rationale, and relevance of reviewing LncRNA/microRNA pairs in the context of ovarian cancer is not that strong. An enhanced introduction to this topic targeting non-specialists will strongly help the reader.

Reply: Thanks, you very much to reviewer for pointed-out this critical issue. We have included new comments explaining the pertinence of study lncRNAs/miRNAs pairs at the end of introduction section as follows: “In particular, we highlighted the description of functional interactions between lncRNAs and microRNAs which interact each other in large regulatory networks that better describe the ovarian cancer cells behavior and tumorigenesis mechanisms, as they may influence gene expression at transcriptional and posttranscriptional level. In ovarian cancer a number of individual studies reporting on lncRNAs and microRNAs genomic profiling have been published, but frequently the information remains dispersed in multiple datasets such as Gene Expression Omnibus (GEO) and The Cancer Genome Atlas (TCGA), thus in this review we aim to put together the omics data and the individual reports on lncRNAs/microRNAs pairs published in diverse studies for better reference and understanding for cancer researchers”.

  1. The authors stated that they are reviewing LncRNA/microRNAs, which have at least two papers published, which seems a very arbitrary selection method. Are there other key LcnRNA/microRNA pairs worth mentioning?

Reply. Thanks for the comment, here we have selected lncRNAs that interact with microRNAs and participate in the regulation of cell migration, angiogenesis, resistance to therapy and Warburg effect. In addition, we selected those that at least two relevant published studies supported their interaction, to better support their involvement in ovarian cancer, as well as to not make the review so long. In our opinion these are the more relevant studies in ovarian cancer. We have corrected the sentence as follows: “For clarity, we have included the description of lncRNAs in which, in our opinion, the studies support their relevant interplay with specific microRNAs impacting the development and progression of ovarian cancer.

  1. The review reads like a list of LcnRNA/microRNA pairs and a summary of published manuscripts. I would prefer to see a more concise summary of published work and an expanded discussion on the relevance and potential of targeting these interactions.

Reply. Thanks for the comments, as you suggested in the point 2, we shortened the manuscript making more concise and succinct the description of data. Also, we emphasized the relevance of lncRNAs and microRNAs, making all of them potential targets for therapy.

  1. I would remove the LcnRNA/microRNA in polycystic ovarian syndrome as the manuscript can benefit from focusing solely on ovarian cancer.

Reply. Thank you for your comment. As reviewer 1 also has concerns about this section, we have removed the polycystic ovary syndrome section and better focused the review on ovarian cancer.